# CROSS-ARCHITECTURE DISTILLATION USING BIDIRECTIONAL CMOW EMBEDDINGS

## ABSTRACT

Large pretrained language models (PreLMs) are revolutionizing natural language processing across all benchmarks. However, their sheer size is prohibitive for small laboratories or deployment on mobile devices. Approaches like pruning and distillation reduce the model size but typically retain the same model architecture. In contrast, we explore distilling PreLMs into a different, more efficient architecture CMOW, which embeds each word as a matrix and uses matrix multiplication to encode sequences. We extend the CMOW architecture and its CMOW/CBOW-Hybrid variant with a bidirectional component, per-token representations for distillation during pretraining, and a two-sequence encoding scheme that facilitates downstream tasks on sentence pairs such as natural language inferencing. Our results show that the embedding-based models yield scores comparable to DistilBERT on QQP and RTE, while using only half of its parameters and providing three times faster inference speed. We match or exceed the scores of ELMo, and only fall behind more expensive models on linguistic acceptability. Still, our distilled bidirectional CMOW/CBOW-Hybrid model more than doubles the scores on linguistic acceptability compared to previous cross-architecture distillation approaches. Furthermore, our experiments confirm the positive effects of bidirection and the two-sequence encoding scheme.

## 1 INTRODUCTION

Large pretrained language models (Devlin et al., 2019; Raffel et al., 2020) (PreLMs) have emerged as de-facto standard methods for natural language processing (Wang et al., 2018; 2019). The common strategy is to pretrain models on enormous amounts of unlabeled text before fine-tuning them for downstream tasks. However, the drawback of PreLMs is that the models are becoming larger and larger with up to several billions of parameters (Brown et al., 2020). This comes with high environmental and economic costs (Strubell et al., 2019) and shifts the development and research in the hands of a few global players only (Bommasani et al., 2021, pp. 10-12). Even though a single pretrained model can be reused for multiple downstream tasks, the sheer model size is often prohibitive. The immense resource requirements prevent using those models in small-scale laboratories and on mobile devices, which is tied to privacy concerns (Sanh et al., 2020b).

There is a need for more efficient models or compressed versions of large models to make AI research more inclusive and energy-friendly, while fostering deployment in applications. Reducing the size of PreLMs using knowledge distillation (Hinton et al., 2015) or model compression (Bucila et al., 2006) is an active area of research (Sanh et al., 2020a; Jiao et al., 2020; Sun et al., 2020). Both knowledge distillation and model compression can be described as teacher-student setups. The student is trained to imitate the predictions of the teacher while using less resources. Typically, a large PreLM takes the role of the teacher while the student is a smaller version of the same architecture. Sharing the same architecture between student and teacher enables using dedicated distillation techniques, e. g., aligning the representations of intermediate layers (Sanh et al., 2020a; Sun et al., 2020).

However, using more efficient architectures as student has already shown promising results such as the task-specific distillation approaches by Tang et al. (2019) and Wasserblat et al. (2020). In their works, the student models are LSTMs (Hochreiter & Schmidhuber, 1997) or models based on a continuous bag-of-words representation (CBOW) (Collobert & Weston, 2008; Mikolov et al., 2013). On the one hand, LSTMs are difficult to parallelize as they need at least $\mathcal{O}(n)$ sequential steps to

encode a sequence of length $n$. On the other hand, CBOW-based models are *not order-aware*, i. e., cannot distinguish sentences with the same words but in different order ("cat eats mouse" vs. "mouse eats cat" are treated equivalent). There are, however, more efficient models such as Mai et al. (2019)'s continual multiplication of words (CMOW) that *do capture word order* by representing each token as a matrix, instead of a vector as in CBOW. A sequence in CMOW is modeled by the non-commutative matrix multiplication, which makes the encoding of a sequence becomes dependent on the word order. We denote such models as *matrix embeddings*.

The present work investigates how order-aware matrix embeddings can be used as student models in cross-architecture distillation from large PreLM teachers. Thus, we complement the existing body of works that focused predominantly on same-architecture distillation. All previous cross-architecture approaches are task-specific, whereas we also explore general distillation. We aim to understand to which extent order-aware embeddings are suited to capture the teacher signal of a large PreLM such as BERT. To this end, we extend Mai et al. (2019)'s CMOW/CBOW-Hybrid model, which is a hybrid variant unifying the strength of CBOW and CMOW, with a bidirectional representation of the sequences. Furthermore, we add the ability to emit per-token representations to facilitate using a modern masked language model objective (Devlin et al., 2019). We investigate both *general distillation*, i. e., the distillation is applied during pretraining on unlabeled text, as well as *task-specific distillation*, when an already fine-tuned PreLM is distilled on a per-task basis. We further introduce a two-sentence encoding scheme to CMOW such that it can deal with similarity and natural language inferencing tasks. This improves the performance by 32% compared to a naive joint encoding.

Our results show that large PreLMs can be distilled into efficient order-aware embedding models and achieve performance comparable to ELMo (Peters et al., 2018) on the GLUE benchmark. On certain tasks, embedding-based models even challenge other size-reduced BERT models such as DistillBERT. In summary, our contributions are:

- We extend order-aware embedding models with bidirection and make them amenable for masked language model pretraining.
- We explore using order-aware embedding models as student models in a cross-architecture distillation setup with BERT as teacher and compare general and task-specific distillation.
- We introduce the first encoding scheme that enables CMOW/CBOW-Hybrid to deal with two-sentence tasks (32% increase over the naive approach).
- Our results show that the best distilled embedding models can be on-par with more expensive models such as ELMo or DistilBERT on certain tasks of the GLUE benchmark, while having a much higher encoding speed (thrice as high as DistilBERT).

Below, we introduce our embedding models, our cross-architecture distillation setup, and our two-sequence encoding scheme. The experimental procedure is described in Section 3. The results are reported in Section 4 and discussed in Section 5, where we also relate our work to the literature.

## 2 METHODS

First, we introduce our bidirectional extension to the CMOW/CBOW-Hybrid model. Subsequently, we introduce our approach for cross-architecture distillation that we use during the pretraining and fine-tuning stages. Finally, we introduce a two-sentence encoding scheme for order-aware embedding models that is crucial for fine-tuning on downstream tasks with paired sentences.

### 2.1 EXTENDING THE ORDER-AWARE EMBEDDING MODELS

We extend the CMOW/CBOW-Hybrid embeddings of Mai et al. (2019) with bidirection and the ability to emit per-token representations as a foundation for cross-architecture distillation. CMOW/CBOW-Hybrid embeddings, **our baseline model**, are a combination of matrix embeddings and vector embeddings. Compared to vector-only embeddings, the word order can be captured because matrix multiplication is non-commutative. Given a sequence of $n$ tokens, matrix-space embeddings $\boldsymbol{X}_j \in \mathbb{R}^{d \times d}$ for each different token $j$, and vector-space embeddings $\boldsymbol{x}_j \in \mathbb{R}^{d_{\text{vec}}}$, the CMOW/CBOW-Hybrid embedding of a sequence of length $n$ is the multiplication of embedding matrices $\boldsymbol{X}_i$

concatenated (symbol $\cdot||\cdot$) to the summation of the embedding vectors $\boldsymbol{x}_i$:

$$\boldsymbol{H}^{(\text{CMOW})} := \boldsymbol{X}_1^{(\text{CMOW})} \cdot \boldsymbol{X}_2^{(\text{CMOW})} \cdots \boldsymbol{X}_n^{(\text{CMOW})}$$

$$\boldsymbol{h}^{(\text{CBOW})} := \sum_{1 \le j \le n} \boldsymbol{x}_j^{(\text{CBOW})}$$

$$\boldsymbol{h}^{(\text{Hybrid})} := \text{flatten}\left(\boldsymbol{H}^{(\text{CMOW})}\right) || \boldsymbol{h}^{(\text{CBOW})}$$

where flatten collapses the matrix into a vector. The original work on CMOW (Mai et al., 2019) has extensively analyzed the CMOW and CBOW components individually and found that joint training is generally preferable. The CMOW and CBOW components can have different dimensionalities, as they are combined via concatenation. We initialize each matrix $\boldsymbol{X}_j$ as identity plus Gaussian noise $\boldsymbol{I}_d + \mathcal{N}(0, \sigma_{\text{init}}^2)$ with $\sigma_{\text{init}} = 0.01$.

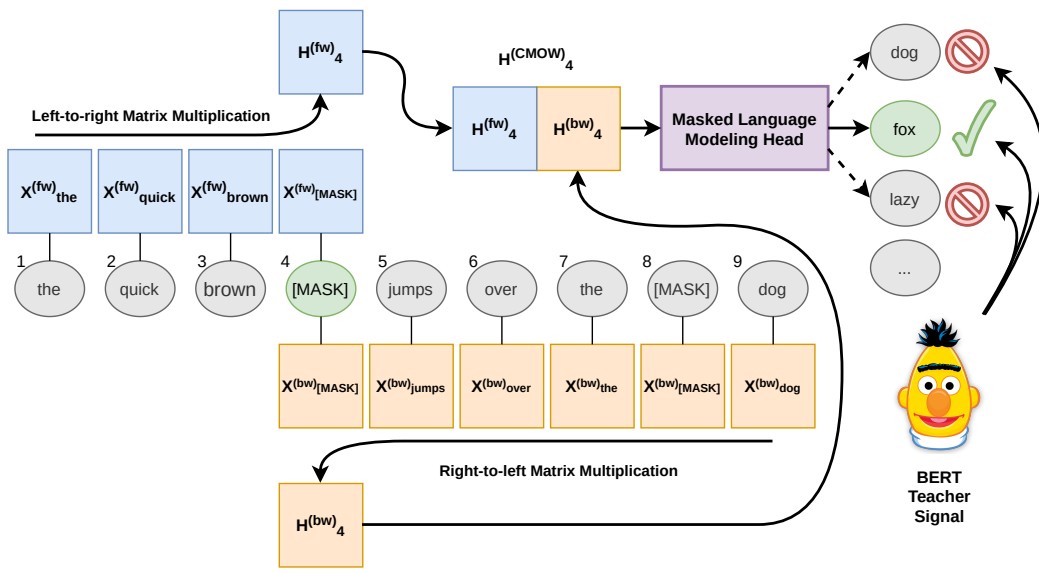

Figure 1: The bidirectional CMOW component of our proposed architecture during pretraining. In this example, the model predicts the masked token at position 4 by concatenating forward and backward matrix embeddings, which are then fed into a masked language modeling head.

**Proposed Model: Bidirectional CMOW/CBOW-Hybrid**  Inspired by the success of bidirection in RNNs (Schuster & Paliwal, 1997), LSTMs (Peters et al., 2018), and Transformers (Devlin et al., 2019), we extend CMOW by a bidirectional component. Hence, we introduce a second set of matrix-space embeddings that are multiplied in reverse order. We then have one matrix embedding for the forward direction $\mathbf{X}^{(\text{fw})} \in \mathbb{R}^{n_{\text{vocab}} \times d \times d}$ and one for the backward direction $\mathbf{X}^{(\text{bw})} \in \mathbb{R}^{n_{\text{vocab}} \times d \times d}$. Then, we concatenate forward and backward directions. Figure 1 illustrates bidirectional CMOW.

Furthermore, we emit one representation per token position $i$, which allows training with a masked language model objective (Devlin et al., 2019). Thus, we are able to make use of the BERT teacher signal for pretraining. Since we can reuse computations, $\mathcal{O}(n)$ matrix multiplications are sufficient to encode a sequence of length $n$. For these intermediate representations, we also modify the CBOW component in a way that it yields partial sums for the forward and backward directions. Formally, we compute the CMOW/CBOW-Hybrid representation as follows:

$$\boldsymbol{H}_i^{(\text{Bidi. CMOW})} := \boldsymbol{X}_1^{(\text{fw})} \cdot \boldsymbol{X}_2^{(\text{fw})} \cdots \boldsymbol{X}_i^{(\text{fw})} || \boldsymbol{X}_n^{(\text{bw})} \cdot \boldsymbol{X}_{n-1}^{(\text{bw})} \cdots \boldsymbol{X}_i^{(\text{bw})}$$

$$\boldsymbol{h}_i^{(\text{Bidi. CBOW})} := \sum_{j=1}^{i} \boldsymbol{x}_j^{(\text{CBOW})} || \sum_{j=i}^{n} \boldsymbol{x}_j^{(\text{CBOW})}$$

$$\boldsymbol{h}_i^{(\text{Bidi. Hybrid})} := \text{flatten}\left(\boldsymbol{H}_i^{(\text{Bidi. CMOW})}\right) || \boldsymbol{h}_i^{(\text{Bidi. CBOW})}$$

For fine-tuning on tasks with full sentences as input, e. g., natural language inferencing, we do not need per-token representations. In this case, we compute the representation of a token sequence as:

$$\boldsymbol{H}^{(\text{Bidi. CMOW})} := \boldsymbol{X}_1^{(\text{fw})} \cdot \boldsymbol{X}_2^{(\text{fw})} \cdots \boldsymbol{X}_n^{(\text{fw})} \parallel \boldsymbol{X}_n^{(\text{bw})} \cdot \boldsymbol{X}_{n-1}^{(\text{bw})} \cdots \boldsymbol{X}_1^{(\text{bw})}$$

$$\boldsymbol{h}^{(\text{CBOW})} := \sum_{j=1}^{n} \boldsymbol{x}_j^{(\text{CBOW})}$$

$$\boldsymbol{h}^{(\text{Bidi. Hybrid})} := \text{flatten}\left(\boldsymbol{H}_i^{(\text{Bidi. CMOW})}\right) \parallel \boldsymbol{h}_i^{(\text{CBOW})}$$

Note that the forward and backward directions of the embedding vectors $\boldsymbol{h}^{(\text{CBOW})}$ conflate to equivalent formulas when we encode entire sequences. Thus, we only need to include a single CBOW representation along with the two CMOW components that do yield different results for the forward and backward direction. At inference time, the model is parallelizable along the sequential dimension.

For regularization, we apply a mild dropout (p=0.1) on both the embeddings and their aggregated representations during pretraining. Then, we feed them into a linear masked language modeling head, see Figure 1, or an MLP classification head to tackle the downstream tasks. We have also experimented with linear, LSTM, and CNN classifiers, which are described in Appendix A. We chose an MLP because it adds nonlinearity to the model without introducing further complexity. The MLP has led to the best average performance across all GLUE tasks (we report the results in Appendix C).

## 2.2 CROSS-ARCHITECTURE DISTILLATION

A central question of our research is whether we can distill a large PreLM, e. g., BERT, into more efficient, non-transformer architectures such as the proposed bidirectional CMOW/CBOW-Hybrid model. This requires a cross-architecture distillation approach, which we describe below.

In general, the idea of knowledge distillation is to compress the knowledge of a large teacher model into a smaller student model (Hinton et al., 2015; Bucila et al., 2006) . It involves a loss function $\mathcal{L}$ that is a combination of two loss terms, i. e., $\mathcal{L} = \alpha \cdot \mathcal{L}_{\text{hard}} + (1 - \alpha) \cdot \mathcal{L}_{\text{soft}}$ with weighting parameter $\alpha$. $\mathcal{L}_{\text{hard}}$ denotes the cross-entropy loss with respect to the ground truth and $\mathcal{L}_{\text{soft}} = \Sigma_i t_i \cdot \log(s_i)$ is the cross-entropy between student logits $\boldsymbol{s}$ and the teacher signal $\boldsymbol{t}$. Optionally, the softmax within $\mathcal{L}_{\text{soft}}$ is flattened by a temperature parameter $T$. We distinguish *general distillation*, where BERT's teacher signal is only used during pretraining, and *task-specific distillation*, where the BERT teacher signal is used during fine-tuning for the downstream (see related work).

Considering our goal to design a cross-architecture distillation, the general distillation approach has the conceptual benefit that the teacher model is not needed for fine-tuning. Thus, the student model is capable of tackling downstream tasks without supervision of the large teacher. This has the benefit that one does not need to carry around the BERT model for adaption to every new downstream task. Above, we have introduced the ability to emit per-token representations with lightweight (bidirectional) CMOW/CBOW-Hybrid embedding models. This enables us now to use BERT's teacher signal during pretraining together with a masked language modeling objective. With other words, this enables us to perform cross-architecture distillation with matrix embeddings.

We consider three variants of cross-architecture distillation in our experiments: a) When using general distillation, depicted in Figure 1, BERT acts as a teacher during pretraining and the model is fine-tuned to downstream tasks on its own. b) For task-specific distillation, BERT acts as a teacher during fine-tuning as shown in Figure 2. For this case, we have the option of either b1) starting with pretrained embeddings (from general distillation, i. e., the a) variant), or b2) starting from scratch with randomly initialized embeddings.

## 2.3 TWO-SEQUENCE ENCODING WITH MATRIX EMBEDDINGS

When fine-tuning our matrix embeddings to downstream tasks, we can deviate from BERT's input processing, even if BERT is used as a teacher. This is because the distillation loss is computed per sentence (pair) and not per token. The input processing of BERT encodes two sequences by joining them into one sequence. For example, in a natural language inferencing task, there is a sentence $A$ that potentially entails a sentence $B$, which is encoded as one sequence using a special separator token.

This encoding scheme is less useful to our matrix embeddings without any attention component since the order-aware matrix multiplications would blend the representation of the two sequences.

To develop an appropriate two-sequence encoding scheme for matrix embeddings, we take inspiration from the pre-transformer era, e. g., Mou et al. (2016), and from SentenceBERT (Reimers & Gurevych, 2019). The key idea is to encode two sentences $A$ and $B$ separately before combining them. As combination operation, we use the absolute elementwise difference and concatenate it to the representations of $A$ and $B$, which we denote as DiffCat:

$$\boldsymbol{h}^{(\text{DiffCat})} = \boldsymbol{h}^{(A)} \parallel |\boldsymbol{h}^{(A)} - \boldsymbol{h}^{(B)}| \parallel \boldsymbol{h}^{(B)}$$

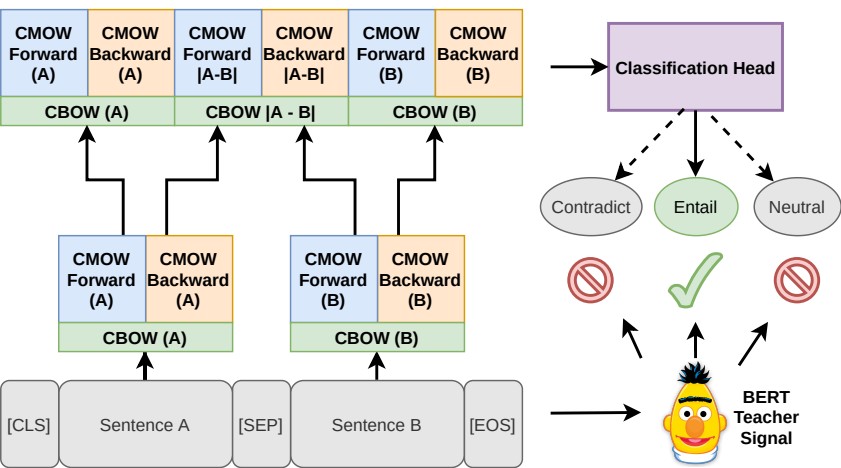

Figure 2: Seperate encoding (DiffCat) for sequence pairs using a Bidirectional CMOW/CBOW-Hybrid model during fine-tuning, optionally, with task-specific distillation with a BERT teacher.

We illustrate this separate encoding scheme during task-specific distillation in Figure 2. The rationale for using a concatenation of both sequence representations along with their difference is that we add a component for the similarity of the two sequence representations, without loss of expressive power.

## 3 EXPERIMENTAL PROCEDURE

The experimental procedure is divided in pretraining on unlabeled text and fine-tuning on the downstream tasks. We provide the details for these two stages and close the section by outlining the downstream tasks and evaluation measures.

### 3.1 PRETRAINING AND GENERAL DISTILLATION

In the pretraining stage, as shown in Figure 1, we train our proposed bidirectional CMOW/CBOW-Hybrid model with a masked language modeling objective (MLM) (Devlin et al., 2019) on large amounts of unlabeled text. The MLM objective is to predict left-out words from their context. We put equal weights on the MLM objective and the teacher signal from BERT ($\alpha = 0.5$). As suggested by Liu et al. (2019) and Sanh et al. (2020a), we do not use the next-sentence-prediction objective of BERT, but only the MLM objective.

As datasets for pretraining, we use a combination of English Wikipedia and Toronto Books (Zhu et al., 2015), as used in original BERT. To reduce the environmental footprint of our experiments, we have only pretrained a single bidirectional CMOW/CBOW-Hybrid model with BERT-base as a teacher on the full unlabeled training data, after pre-experiments on $10\%$ of the training data showed that the selected CMOW/CBOW-Hybrid with distillation exceeded the performance of the baseline.

We use matrix embeddings of size $20 \times 20 = 400$ ($d = 20$) for both the CMOW directions (forward and backward) and vector embeddings of size $d_{\text{vec}} = 400$. We use BERT's tokenizer and its vocabulary for both, the teacher and student. The BERT tokenizer relies primarily on the WordPiece algorithm (Wu et al., 2016), which yields a high coverage while maintaining a small vocabulary.

## 3.2 Fine-tuning and Task-specific Distillation

In the fine-tuning stage, as shown in Figure 2, the pretrained model is adapted for each downstream task individually. The training objective for fine-tuning is either cross-entropy with the ground truth (in general distillation) or a mixture of the ground truth loss and cross-entropy with respect to the teacher's logits (in task-specific distillation). Again we put equal weight on ground truth and teacher signal along ($\alpha = 0.5$) with unit temperature. To facilitate distillation on regression tasks, we follow Raffel et al. (2020) and cast STS-B from regression to classification by binning the scores into intervals of 0.2. To encode the inputs for two-sequence tasks, we use a sequential encoding similar to BERT and the proposed DiffCat encoding (see Section 2.3).

For task-specific distillation, we employ an uncased BERT-base model[1] from the Huggingface repository that has already been fine-tuned for each task of the GLUE benchmark. We have fine-tuned the BERT model ourselves on the tasks STS-B, where we applied binning, and MNLI, since the Huggingface model led to subpar results. We use the same fine-tuned BERT model as a teacher for all reported results with task-specific distillation.

We seek a fair comparison between unidirectional CMOW/CBOW-Hybrid baseline model and our bidirectional model. As such, we allow both models to equally benefit from the BERT's teacher signal during fine-tuning. For this comparison, we use random initialization for both models because the Mai et al. (2019)'s pretrained embeddings would come with a different vocabulary that covers only 53% of the one of BERT. Throughout the other experiments, we initialize our bidirectional CMOW/CBOW-Hybrid with the pretrained embeddings from general distillation, while we isolate the effect of task-specific distillation in a dedicated experiment.

For hyperparameter optimization, we tune learning rates in the range $[10^{-3}, 10^{-6}]$. In total, we have conducted 306 training and evaluation runs for hyperparameter optimization of each GLUE task. To determine the best model, we use each task's evaluation measure on the development set. We run each model for 20 epochs with early stopping (5 epochs patience). We select appropriate batch sizes on the basis of preliminary experiments and training data sizes. For a more detailed discussion, we refer to Appendix B. The hyperparameters of the best-performing models are reported in Appendix C.

We have further experimented with data augmentation and using exclusively the teacher signal during task-specific distillation ($\alpha = 1$). In some tasks, we could further increase the results by a small margin, but found no consistent improvement. These results are also reported in Appendix C.

## 3.3 Downstream Tasks and Measures

We use the GLUE benchmark (Wang et al., 2018) to evaluate our models. The GLUE benchmark consists of nine tasks for English language comprehension (Wang et al., 2018). These tasks comprise natural language inference (MNLI-m, QNLI, WNLI, RTE), sentence similarity (QQP, STS-B, MRPC), linguistic acceptibility (CoLA), and sentiment analysis (SST-2). All tasks are based on pairs of sentences except for CoLA and SST-2, which are single-sentence tasks. The GLUE benchmark explicitly encourages using different fine-tuning strategies for different tasks. For our evaluation, we use the GLUE development set along with its task-specific measures. As such, the performance on all four NLI tasks as well as SST-2 is measured with accuracy. CoLA is evaluated by Matthews correlation coefficient. Similarity tasks are measured by the average of Pearson and Spearman correlation for the STS-B task, and as the average of accuracy and $F_1$-score for MRPC and QQP.

## 4 Results

We present the results along the design choices introduced in Section 2, namely bidirection, cross-architecture distillation approaches, and two-sequence encoding scheme. Finally, we compare our best embedding methods with ELMo and BERT distillates from the literature. For brevity, we focus on reporting key results. The detailed results can be found in Appendix C.

**Bidirectional CMOW/CBOW-Hybrid versus Baseline**   To isolate the effect of the bidirectional component, we compare unidirectional CMOW/CBOW-Hybrid with bidirectional CMOW/CBOW-

---

[1]https://huggingface.co/textattack

Hybrid under equal conditions. We train both variants from scratch for the downstream tasks, while using a BERT's teacher signal. Table 1 shows the results of comparing unidirectional Hybrid embeddings with the proposed bidirectional Hybrid embeddings. Bidirection helps on the tasks MNLI, MRPC, QNLI, SST-2, STS-B, and WNLI. On the other tasks, the difference is marginal. We have an average improvement of $1\%$ of the bidirectional model over the unidirectional model across all tasks of the GLUE benchmark.

Table 1: Comparison of bidirectional CMOW/CBOW-Hybrid versus unidirectional and bag-of-words baselines under task-specific distillation with DiffCat encoding and MLP classifier.

| Model Type | Score | CoLA | MNLI-m | MRPC | QNLI | QQP | RTE | SST-2 | STS-B | WNLI |
|---|---|---|---|---|---|---|---|---|---|---|
| Hybrid, rand. init. | 62.5 | **13.1** | 62.5 | 74.3 | 71.5 | **86.6** | **58.1** | 83.1 | 58.6 | 56.3 |
| Bidirectional Hybrid, rand. init. | **63.2** | 13.0 | **63.3** | **75.7** | **72.6** | 86.1 | 57.4 | **83.3** | **59.7** | **57.7** |

**General Distillation vs. Task-specific Distillation**  Next, we compare general distillation with task-specific distillation. As shown in Table 2, using general distillation leads to better results for five tasks (MNLI, MRPC, QQP, STS-B, and RTE) compared to task-specific distillation. For the other four tasks (CoLA, QNLI, SST-2, and WNLI), task-specific distillation achieves higher scores. The average score of general distillation is higher than with task-specific distillation in both pretrained and randomly initialized cases.

Table 2: Comparison of task-specific vs. general distillation using bidirectional CMOW/CBOW-Hybrid embeddings and MLP classifier. In 5 out of 9 tasks, general distillation performs best.

| Distillation Type | Score | CoLA | MNLI-m | MRPC | QNLI | QQP | RTE | SST-2 | STS-B | WNLI |
|---|---|---|---|---|---|---|---|---|---|---|
| General | **66.6** | 16.7 | **66.6** | **79.7** | 71.7 | **87.2** | **61.0** | 82.9 | **76.9** | 56.3 |
| Task-specific, rand. init | 63.2 | 13.0 | 63.3 | 75.7 | **72.6** | 86.1 | 57.4 | **83.3** | 59.7 | **57.7** |
| Task-specific, pretrained | 64.6 | **23.3** | 61.8 | 75.0 | 72.0 | 86.3 | 59.9 | 82.9 | 62.9 | **57.7** |

**DiffCat Encoding vs. Joint Encoding**  Now, we compare the two encoding schemes, whose results are presented in Table 3. The best scores were achieved with DiffCat encoding. Most remarkable is the improvement from $18.1$ to $56.2$ on the sentence similarity task STS-B when encoding the sentence pair input via DiffCat. The average improvement of DiffCat encoding is $32\%$ across the two-sentence GLUE tasks.

Table 3: Comparison of DiffCat encoding vs. joint BERT-like encoding. Both variants use pretrained bidirectional CMOW/CBOW-Hybrid embeddings with MLP under task-specific distillation. DiffCat encoding improves the results in all cases except for RTE with the largest margin on STS-B.

| Two-Sentence Encoding | MNLI-m | MRPC | QNLI | QQP | RTE | STS-B | WNLI |
|---|---|---|---|---|---|---|---|
| BERT-like Encoding | 47.4 | 71.6 | 60.0 | 79.5 | 57.8 | 18.1 | 56.3 |
| DiffCat Encoding | 62.6 | 74.5 | 68.6 | 85.7 | 56.3 | 56.2 | 56.3 |

**Comparing bidirectional CMOW/CBOW-Hybrid to the Literature**  Finally, Table 4 shows the results of the best-performing bidirectional CMOW/CBOW-Hybrid variants using any of the three considered distillation methods. As described by Wasserblat et al. (2020), a model needs to capture context and linguistic structure to perform well on CoLA. We doubled the results for CoLA and SST-2 compared to the best previously reported cross-architecture distillation approaches by Wasserblat et al. (2020). Our best models scored higher than ELMo (Peters et al., 2018) on the tasks MRPC, QNLI, QQP, RTE, and WNLI. We achieve higher scores than DistilBERT on RTE and WNLI.

**Runtime Performance and Parameter Count**  To compare runtime, we generate 1,024 batches with 256 random sequences of length 64 and measure the time to encode the sequences with gradient computation disabled. As shown in Table 5, CMOW/CBOW-Hybrid is more than 3 times faster than the fastest competitor, DistilBERT, and only uses about half of its parameters. The inference speed of CMOW/CBOW-Hybrid could be increased even further because the $\mathcal{O}(n)$ steps to encode a sequence of length $n$ can be parallelized to $\mathcal{O}(\log n)$ *sequential* steps, since matrix multiplication is associative.

Table 4: Comparison of best embedding-based methods (in bold) with methods from the literature on the validation set of the GLUE benchmark

| Method | Score | CoLA | MNLI-m | MRPC | QNLI | QQP | RTE | SST-2 | STS-B | WNLI |
|---|---|---|---|---|---|---|---|---|---|---|
| ELMo (Peters et al., 2018) | 68.7 | 44.1 | 68.6 | 76.6 | 71.1 | 86.2 | 53.4 | 91.5 | 70.4 | 56.3 |
| DistilBERT (Sanh et al., 2020a) | 77.0 | 51.3 | 82.2 | 87.5 | 89.2 | 88.5 | 59.9 | 91.3 | 86.9 | 56.3 |
| MobileBERT (Sun et al., 2020) | — | 51.1 | 84.3 | 88.8 | 91.6 | 70.5 | 70.4 | 92.6 | 84.8 | — |
| CBOW (Wasserblat et al., 2020) | — | 10.0 | — | — | — | — | — | 79.1 | — | — |
| BiLSTM (Wasserblat et al., 2020) | — | 10.0 | — | — | — | — | — | 80.7 | — | — |
| Hybrid (Mai et al., 2019) | — | — | — | — | — | — | — | 79.6 | 63.4 | — |
| Word2rate (Phua et al., 2021) | — | — | — | — | — | — | — | 65.7 | 53.1 | — |
| **Bidi. Hybrid + MLP (ours)** | **68.0** | **23.3** | **66.6** | **80.9** | **72.6** | **87.2** | **61.0** | **84.0** | **76.9** | **59.2** |

Table 5: Number of parameters and inference time of the models. Inference time is measured as encoding speed without gradient computation on an NVIDIA A100-SXM4-40GB card

| Model | # Parameters | Encoding speed (sentences / second) |
|---|---|---|
| ELMo | 94M | 1.1k |
| BERT-base | 109M | 4.6k |
| DistilBERT-base | 66M | 9.2k |
| MobileBERT | 25M | 5.5k |
| TinyBERT (4 layer) | **14M** | **30.0k** |
| Bidi. CMOW/CBOW-Hybrid | 37M | **30.0k** |

## 5 DISCUSSION AND RELATED WORK

**Key Results** We have shown that BERT can be distilled into efficient matrix embedding models during pretraining by emitting intermediate representations. We have further introduced a bidirectional component and a separate two-sequence encoding scheme for CMOW-style models. We have observed that the general distillation approach, i.e., using the BERT teacher only during pretraining, leads to results that are oftentimes even better than those achieved with task-specific distillation. This is an interesting result because all previous works on cross-architecture distillation relied on task-specific distillation. Our proposed model offers an encoding speed at inference time that is three times faster than DistilBERT and more than five times faster than MobileBERT.

**Limitations** Currently, matrix embeddings still fall behind other BERT distillates, most notably, MobileBERT, on many of the downstream tasks. In particular, detecting linguistic acceptability remains a challenge for non-transformer methods, even though we improve upon previous cross-architecture distillation approaches. So far, we have not analyzed the trade-off between embedding size and downstream performance. We rely on general arguments for the benefits of increased dimensionality (Wieting & Kiela, 2019).

**Reflection w.r.t. Distillation and Size-reduction Approaches** In general distillation, a PreLM is distilled into a student model during pretraining. DistilBERT (Sanh et al., 2020a) is such a general-purpose language model that has been distilled from BERT. Apart from masked language modeling and distillation objectives, the authors also introduce a cosine loss term to align the student's and teacher's hidden states (layer transfer). Furthermore, the student is initialized with selected layers of the teacher. MobileBERT (Sun et al., 2020) introduced a bottleneck to BERT such that layers can be transferred to student models with smaller dimensions.

In task-specific distillation, the teacher signal is used during fine-tuning. Sun et al. (2019) use layer-wise distillation objectives and initialize with teacher weights to train BERT students with fewer layers. TinyBERT (Jiao et al., 2020) applies knowledge distillation in both stages, pretraining and fine-tuning. TinyBERT (Jiao et al., 2020) yields high throughput rates comparable to CMOW/CBOW-Hybrid. Yet, TinyBERT requires keeping the teacher model for fine-tuning. We have shown that CMOW/CBOW-Hybrid is even better with general distillation only than under task-specific distillation. TinyBERT also augments the training data, which we have also considered but found no consistent improvement. Turc et al. (2019) analyzes the interaction between pre-training and fine-tuning with BERT models and find that pretrained distillation works well, which agrees with our findings on the importance of pretraining with CMOW-style models. LadaBERT (Mao et al., 2020)

combines knowledge distillation with pruning and matrix factorization. Other approaches consider distillation in multi-lingual (Tsai et al., 2019) or multi-task settings (Yang et al., 2019).

The works described above assume that the teacher and student share the same architecture. However, the student model does not need to have the same architecture as the teacher, what we then call cross-architecture distillation. Wasserblat et al. (2020) use a simple feed forward network with CBOW embeddings and a bidirectional LSTM model as students. Both models perform well in several downstream tasks. Tang et al. (2019) explore distilling BERT into a single layer BiLSTM without using any additional training data or modifications to the teacher architecture. Their distillation-based approach yields improvements compared to a plain BiLSTM without teacher signal: about 4 points on all reported tasks (QQP, MNLI, and SST-2). This has motivated us to investigate whether even simpler models can be used as students of a BERT teacher.

Other techniques for reducing the size of a model are pruning and quantization. Pruning approaches such as Sanh et al. (2020b) reduce the number of parameters. Still, the resulting smaller models use the same architecture as their larger counterparts and, thus, pruning does not necessarily improve inference speed. Quantization is a common post-processing step to reduce model size by decreasing the floating point precision of the weights (Gupta et al., 2015; Wu et al., 2020). Pruning and quantization can be applied in conjunction with knowledge distillation (Sanh et al., 2020b; Sun et al., 2020). Aside from techniques for reducing the model size, there is also a tremendous effort to improve the efficiency of the transformer architecture in the first place (Tay et al., 2020).

The literature focuses on reducing the size of PreLMs via distillation, pruning, and quantization. Specialized techniques depend on teacher and student sharing the same architecture, and thus, are not applicable for cross-architecture distillation. So far, only few recent works consider distilling PreLMs into other architectures like LSTMs and CBOW models. In this work, we show that cross-architecture distillation with order-aware embedding models as students outperform previous cross-architecture distillation approaches and achieve scores comparable to ELMo, while using less computational resources. Finally, all previous cross-architecture distillation approaches are task-specific, while we could show that general distillation can lead to even higher scores than task-specific distillation.

# 6 CONCLUSION AND FUTURE WORK

We have introduced three extensions to the CMOW/CBOW-Hybrid model: a bidirectional component, a separate two-sequence encoding scheme, and the ability to emit per-token representations. These per-token representations allow us to distill BERT into CMOW/CBOW-Hybrid already during pretraining with a masked language modeling objective. Our results show that a separate encoding scheme improves the performance of CMOW/CBOW-Hybrid on two-sentence GLUE tasks by more than 30%, while bidirection improves the performance by 1% compared to the unidirectional model. Furthermore, we have shown that general distillation seems to be sufficient and task-specific distillation is not necessary for most GLUE tasks. In comparison with more expensive models from the literature, our embedding-based approach achieves scores that match or exceed the scores of ELMo and are competitive to DistilBERT on QQP and RTE with only half of its parameters and thrice its encoding speed. While linguistic acceptability remains a challenge for non-transformer models, our approach yields notably higher scores than previous cross-architecture distillation approaches.

In future work, one could further improve the efficiency by applying pruning (Sanh et al., 2020b) and/or quantization (Wu et al., 2020) techniques on the learned matrices to allow sparse matrix multiplication during encoding. Future work could explore what components would be necessary to improve the scores on particularly challenging downstream tasks such as detecting linguistic acceptability. This might include the introduction of a small attention module like the one of gMLP (Liu et al., 2021) to CMOW/CBOW-Hybrid. Since we have shown how to emit (and train) per-token representations with CMOW, an interesting direction of future work would be to explore whether CMOW representations are suited as order-aware embeddings in transformer models, which might alleviate the need for a dedicated position embedding. Similarly, the per-token representations enables tackling question answering tasks with CMOW-style models, in which a span of the context ought to be predicted. Finally, matrix embeddings are technically not limited to natural language processing. Therefore, another interesting path of future work would be to explore the use of matrix embeddings in other domains where sequences of discrete elements need to be encoded, e. g., event sequences, gene sequences, or item sequences in recommender systems.

## ETHICS STATEMENT

There has been criticism about large-scale language models and the amplification of bias through the reduction of model size. This is particularly a problem when the model is not explainable. In our proposed method, we distill large-scale pretrained language models into models that do not have a nonlinearity (except for the final classification head). Since nonlinearities make explainabilty more difficult, we hope to contribute to more explainable language models with our work, even though explainability is not the main focus of the present work.

Another concern with large language models is their immense consumption of resources. In our work, we have only conducted the expensive pretraining step once on the full unlabeled training data. Only afterwards, we conduct multiple "cheap" runs to optimize hyperparameters for downstream tasks. Thus, our experiments themselves have been conducted resource efficient. Furthermore, this research contributes to the development of simpler and more efficient models for natural language processing.

## REPRODUCIBILITY STATEMENT

To ensure that our experiments are reproducible, we have tracked all hyperparameters throughout experimentation. We have described all steps for data processing, i. e., tokenization (we use the BERT tokenizer) and binning (for STS-B) in the main part. The datasets from the GLUE benchmark are well-known to the community, because of which we have omitted a detailed description, except for the metrics that we have reported.

We make the code and pretrained embeddings available upon acceptance of the paper. For now, we provide the code for our experiments as supplementary material.

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

APPENDIX

## A OVERVIEW OF ARCHITECTURAL CHOICES

Figure 3 provides an overview of the architectural choices explored in this paper. We use pretrained BERT Devlin et al. (2019) as well as the embeddings from Mai et al. (2019) as teacher for general distillation. Additionally, we pretrain a model CMOW/CBOW-Hybrid with our extension of masked language model training and bidirection on the same English Wikipedia + Toronto Books dataset. These pretrained embeddings may serve as initialization for the downstream classification models. We also evaluate downstream classification models that have been initialized randomly. For the downstream classification models, we consider three types of embeddings along with four types of classifiers.

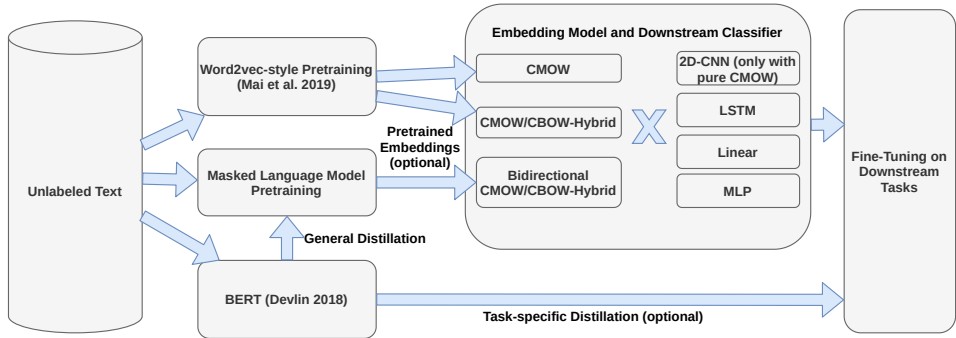

Figure 3: All considered for embeddings and downstream classifiers, pretraining and fine-tuning.

For training on the downstream task, we use once again BERT as a teacher for task-specific distillation, while our experiments on general distillation only benefit from the initalization of the MLM-pretrained CMOW/CBOW-Hybrid model. Throughout the main part of the paper, we have reported scores with an MLP downstream classifier, which achieved the highest average scores. We have further experimented with a linear downstream classifier, LSTM, and 2D-CNN, which we briefly describe below.

**LSTM** We have further experimented with pooling the sequence of embeddings with an LSTM. In the past, BiLSTM models have been successfully used in sentiment analysis tasks Xu et al. (2019); Hameed & Garcia-Zapirain (2020). In an LSTM network, the information at hand is propagated in the forward direction. Thus, each state $t$ depends on its predecessor $t-1$. BiLSTM are LSTM networks, in which the inputs are processed twice: once in the forward direction and once in the backward direction, generating a set of two outputs. In order to generate the output vectors, the output of a single BiLSTM block is fed into an MLP, consisting of two consecutive linear layers with ReLU activation functions. Note that the BiLSTM operates on a sequence of token embeddings, instead of operating on pooled sentence embeddings like the other student models. We apply a dropout of $0.5$ after the first linear layer.

**CNN** We also explore a 2D-CNN classifier that induces a bias for learning two-dimensional structures within the (aggregated) embedding matrices. The CNN consists of one transposed convolution, which increases the matrix dimensions by a factor of four. Following that, we employ a block of three convolutional layers, the first one having a single filter (or two, for hybrid variants) and a kernel size of four, with the remaining two layers having 3 (4) kernels with stride 2. To avoid distorting the input embeddings, no padding is applied. ReLU is used for all activation functions. We apply BatchNorm for regularization before the last convolutional layer's output is flattened and passed into a linear layer, which produces the predictions. We add a dropout of $0.4$ before the last linear layer.

Table 6: Hyperparameter search space and optimization method

| Hyperparameter | Range | Opt. method |
|---|---|---|
| *— General Distillation —* | | |
| Learning rate | $\{10^{-3}, 5 \cdot 10^{-4}, 10^{-4}, 5 \cdot 10^{-5}, 10^{-5}\}$ | grid search |
| Warmup steps | $\{0, 500\}$ | grid search |
| Embedding dropout | $\{0, 0.1\}$ | grid search |
| Hidden unit dropout | $\{0.2\}$ | fixed |
| Batch size | $\{1,8,32,64,128,256\}$ | manual |
| *— Task-specific Distillation —* | | |
| Learning rate | $\{10^{-3}, 5 \cdot 10^{-4}, 10^{-4}, 5 \cdot 10^{-5}, 10^{-5}, 5 \cdot 10^{-6}\}$ | grid search |
| Embedding type | Hybrid, CMOW, CBOW | grid search |
| Embedding initialization | random, pretrained | grid search |
| DiffCat | true, false | grid search |
| Bidirectional | true, false | grid search |
| Classifier | Linear Probe, MLP, CNN, BiLSTM | grid search |

## B  DISCUSSION OF HYPERPARAMETERS AND LOSS FUNCTIONS FOR DISTILLATION

We list hyperparameter search spaces along with their optimization methods in Table 6. For the experiments on data augmentation and using only soft loss, we keep the configurations of the best models (See Table 7 and tune the learning rate, again. We optimize over all six initial learning rates, namely $\{10^{-3}, 5 \cdot 10^{-4}, 10^{-4}, 5 \cdot 10^{-5}, \text{ and } 10^{-5}\}$. All initial learning rates decay linearly over the course of training. Note, we also experimented with using warmup steps versus no warmup for the learning rate schedule. As the warmup did not improve the results, we did not use it.

For the softmax temperature, we find that $T = 1$ is often used Hinton et al. (2015); Mao et al. (2020); Jiao et al. (2020); Mishra & Marr (2018); Polino et al. (2018). Since a higher temperature also flattens the curve over all predictions, it could add too much noise and it is therefore better to use a smaller temperature Chen et al. (2017). Setting the weight $\alpha = 1$ corresponds to only using hard loss and $\alpha = 0$ to only using soft loss. Since we do not want to discard any information stemming from the hard loss, we do not follow the approach of Wasserblat et al. who only use the soft loss Wasserblat et al. (2020) but instead, we employ a vanilla knowledge distillation approach following Hinton et al. Hinton et al. (2015).

Hinton et al. (2015) state, that using cross-entropy loss on the softmax-temperature with a large temperature, for example $T = 20$, corresponds to only using the Mean Square Error (MSE) loss on the raw student and teacher logits. Therefore it is also common to use this loss for the soft distillation loss Tang et al. (2019); Wasserblat et al. (2020); Mukherjee & Awadallah (2020). While Tang et al. Tang et al. (2019) used the weighted hard cross-entropy loss in the overall loss calculation, Wasserblat et al. and Mukherjee et al. only used the soft loss Tang et al. (2019); Wasserblat et al. (2020); Mukherjee & Awadallah (2020). A disadvantage of MSE loss, is that every error has a huge effect on the overall loss, since it is squared. Another point is, that Hinton et al. found it beneficial to use small temperature values if the teacher is way bigger than the student Hinton et al. (2015). Since using MSE loss corresponds to using big $T$ values, this loss does not apply to our use case of using small students for lower bound knowledge distillation, but with cross-entropy loss, we still have the possibility to achieve the behavior of the MSE loss by setting the value of $T$ to a big value.

## C  DETAILED RESULTS

In the following, we provide detailed results for task-specific distillation including the different downstream classifiers, unidirectional CMOW/CBOW-Hybrid, and joint two-sequence encoding. The best performing model per task are marked in bold. We abbreviate CMOW/CBOW-Hybrid as 'Hybrid'.

For the unidirectional baseline model CMOW/CBOW-Hybrid, we initialize with pretrained embeddings provided by Mai et al. (2019)[2], which cover 54% of BERT's vocabulary. As initialization for the newly developed bidirectional CMOW/CBOW-Hybrid models, we use our own pretrained model obtained by general distillation with BERT, as described in Section 3.

Table 7 summarizes the best performing models along with their hyperparameter configurations for each task. Note that we have chosen to use an MLP downstream classifier for the results reported in the main part of this work. Using an MLP downstream classifier has led to the highest average scores across all GLUE tasks.

In Table 8, we report an extended version of the comparison with the literature. Here, we also include our BERT-base teacher model, aswell as TinyBERT Jiao et al. (2020) and Tang et al. (2019)'s distilled BiLSTM. Note that TinyBERT and BiLSTM are not fully comparable, because those numbers are reported on the official GLUE test set, while we have used the validation set for our experiments.

In Table 9, we report the results for all downstream classifiers without the DiffCat aggregation but with a sequential BERT-like two-sequence encoding. It is interesting to see that the CMOW-only variant with 2D-CNN classifier leads to the best scores on sentiment analysis task SST-2. Note that all CMOW variants reported in this table are unidirectional and use task-specific distillation.

In Table 10, we report the results for all downstream classifiers with DiffCat two-sequence encoding. Here we observe, that pretrained CBOW with an MLP classifier leads to the best results on sentence similarity (STS-B). Again, all CMOW variants reported in this table are unidirectional and use task-specific distillation.

In Table 11, we report the results for bidirectional models with DiffCat two-sequence encoding.

From all tables combined, we see that Bidirectional CMOW/CBOW-Hybrid model leads to the highest scores on average, even though, on individual tasks, some other variations of the approach lead to higher scores. Thus, we regard bidirectional CMOW/CBOW-Hybrid as our primary model, whose scores we have reported in the main paper, while isolating the effect of the individual components (bidirection, DiffCat encoding, distillation strategies).

We list the number of parameters in Tables 12 and 13. While the absolute numbers might seem high, it is important to note that we have also counted the parameters of the embeddings. As we show in the tables, the number of parameters in the classification models is much lower.

We have performed further experiments with the best performing model for each task: data augmentation and using only soft loss.

**Using Only Soft Loss**    We study the influence of the alpha value used in the loss function, based on the best results obtained with the initial $\alpha = 0.5$. The goal is to investigate whether using only soft loss, i. e., setting $\alpha = 0.0$ leads to different results. As Table 8 shows, using only soft loss improves only the MRPC task by a small margin.

**Data Augmentation**    We conduct a further study with data augmentation as in TinyBERT (Jiao et al., 2020). We employ their technique of replacing words by similar word embeddings and nearest predictions from BERT to augment the GLUE training datasets. The results are shown in Table 8. We find that the effect of data augmentation is small. An improvement was only observed on SST-2 (+1.2 points) and STS-B (+3.6).

---

[2]Downloaded from Zenodo: `https://zenodo.org/record/3933322#.YKJ_uxKxXJU`

| Task | Score | Classifier | Emb. type | Emb. initialization | DiffCat | Bidirectional | Learning rate |
|------|-------|-----------|-----------|--------------------|---------|--------------|--------------|
| CoLA | 23.3 | MLP | CMOW/CBOW-Hybrid | pretrained | true | true | 1.0E-4 |
| MNLI-m | 63.3 | MLP | CMOW/CBOW-Hybrid | not pretrained | true | true | 1.0E-4 |
| MRPC | 78.2 | MLP | CBOW | pretrained | true | false | 1.0E-3 |
| QNLI | 72.6 | MLP | CMOW/CBOW-Hybrid | not pretrained | true | true | 5.0E-5 |
| QQP | 86.6 | MLP | CMOW/CBOW-Hybrid | not pretrained | true | false | 1.0E-4 |
| RTE | 59.9 | MLP | CMOW/CBOW-Hybrid | pretrained | true | true | 5.0E-4 |
| SST-2 | 86.8 | CNN | CMOW | not pretrained | false | false | 5.0E-4 |
| STS-B | 66.0 | MLP | CBOW | pretrained | true | false | 1.0E-4 |
| WNLI | 69.0 | CNN | CMOW | pretrained | false | false | 1.0E-5 |

Table 7: Hyperparameter configurations for best-performing models by GLUE task

Table 8: Scores on the GLUE development set. Our best performing general distillation and task-specific distillation models are highlighted in bold font per task. References indicate sources of scores. The ⋆-symbol indicates numbers on the official GLUE test set. CMOW/CBOW-Hybrid is abbreviated as 'Hybrid'.

| | Score | CoLA | MNLI-m | MRPC | QNLI | QQP | RTE | SST-2 | STS-B | WNLI |
|---|-------|------|--------|------|------|-----|-----|-------|-------|------|
| *— large-scale pre-trained language models —* | | | | | | | | | | |
| ELMo (Sanh et al., 2020a) | 68.7 | 44.1 | 68.6 | 76.6 | 71.1 | 86.2 | 53.4 | 91.5 | 70.4 | 56.3 |
| BERT-base (Sanh et al., 2020a) | 79.5 | 56.3 | 86.7 | 88.6 | 91.7 | 89.6 | 69.3 | 92.7 | 89.0 | 53.5 |
| BERT-base (our teacher model) | 78.9 | 57.9 | 84.2 | 84.6 | 91.4 | 89.7 | 67.9 | 91.7 | 88.0 | 54.9 |
| Word2rate Hybrid (Phua et al., 2021) | — | — | — | — | — | — | — | 65.7 | 53.1 | — |
| *— general distillation baselines —* | | | | | | | | | | |
| DistilBERT (Sanh et al., 2020a) | 77.0 | 51.3 | 82.2 | 87.5 | 89.2 | 88.5 | 59.9 | 91.3 | 86.9 | 56.3 |
| MobileBERT (Sun et al., 2020) | — | 51.1 | 84.3 | 88.8 | 91.6 | 70.5 | 70.4 | 92.6 | 84.8 | — |
| *— task-specific distillation baselines —* | | | | | | | | | | |
| ⋆TinyBERT (Jiao et al., 2020) | — | 54.0 | 84.5 | 90.6 | 91.1 | 88.0 | 70.4 | 93.0 | 90.1 | — |
| ⋆BiLSTM (Tang et al., 2019) | — | — | 73.0 | — | 78.2 | — | — | 90.7 | | — |
| CBOW-FFN (Wasserblat et al., 2020) | — | 10.0 | — | — | — | — | — | 79.1 | — | — |
| BiLSTM (Wasserblat et al., 2020) | — | 10.0 | — | — | — | — | — | 80.7 | — | — |
| *— general distillation (ours) —* | | | | | | | | | | |
| Bidi. Hybrid + Linear | 65.1 | 15.0 | 63.6 | **80.9** | 70.7 | 84.3 | 56.7 | **84.0** | 71.1 | **59.2** |
| Bidi. Hybrid + MLP | 66.6 | **16.7** | 66.6 | 79.7 | **71.7** | **87.2** | **61.0** | 82.9 | **76.9** | 56.3 |
| *— task-specific distillation (ours) —* | | | | | | | | | | |
| CMOW + CNN (rand. init.) | 54.6 | 13.4 | 45.6 | 72.3 | 61.2 | 82.6 | 56.3 | **86.8** | 15.0 | 57.8 |
| CMOW + CNN (pretrained) | 56.2 | 18.3 | 50.1 | 71.8 | 60.5 | 80.6 | 57.0 | 85.0 | 13.2 | **69.0** |
| CBOW + MLP (pretrained) | 63.8 | 14.0 | 61.7 | **78.2** | 70.8 | 86.2 | 57.4 | 83.8 | **66.0** | 56.3 |
| Hybrid + MLP (rand. init.) | 62.5 | 13.1 | 62.5 | 74.3 | 71.5 | **86.6** | 58.1 | 83.1 | 58.6 | 56.3 |
| Bidi. Hybrid + MLP (rand. init.) | 63.2 | 13.0 | **63.3** | 75.7 | **72.6** | 86.1 | 57.4 | 83.3 | 59.7 | 57.7 |
| Bidi. Hybrid + MLP (pretrained) | 64.6 | **23.3** | 61.8 | 75.0 | 72.0 | 86.3 | **59.9** | 82.9 | 62.9 | 57.7 |
| *— further experiments on best-performing task-specific distillation models —* | | | | | | | | | | |
| Only soft loss ($\alpha = 0$) | 64.0 | 19.9 | 62.3 | 78.7 | 72.4 | 68.5 | 56.3 | 86.6 | 62.4 | 69.0 |
| Data augmentation | 63.5 | 21.2 | 47.3 | 76.2 | 72.1 | 86.6 | 52.7 | 88.0 | 69.6 | 57.7 |

Table 9: Scores on the GLUE development set without DiffCat encoding

| Task-Specific Distillation | Score | CoLA | MNLI-m | MRPC | QNLI | QQP | RTE | SST-2 | STS-B | WNLI |
|---|---|---|---|---|---|---|---|---|---|---|
| *— task-specific finetuning (ours) —* | | | | | | | | | | |
| Teacher BERT-base | 78.9 | 57.9 | 84.2 | 84.6 | 91.4 | 89.7 | 67.9 | 91.7 | 88.0 | 54.9 |
| *— task-specific distillation (ours) CBOW not pretrained —* | | | | | | | | | | |
| Linear probe | 52.8 | 12.2 | 43.0 | 72.3 | 60.1 | 74.8 | 55.6 | 82.8 | 17.7 | 56.3 |
| MLP | 53.2 | 13.0 | 46.3 | 71.3 | 59.7 | 76.9 | 54.5 | 82.9 | 17.5 | 56.3 |
| CNN | 52.8 | 11.7 | 43.0 | 72.1 | 60.1 | 77.5 | 54.5 | 82.7 | 17.2 | 56.3 |
| BiLSTM | 52.1 | 10.9 | 44.9 | 70.8 | 59.8 | 78.1 | 54.5 | 81.3 | 12.3 | 56.3 |
| *— task-specific distillation (ours) CBOW pretrained —* | | | | | | | | | | |
| Linear probe | 52.4 | 11.0 | 43.2 | 72.1 | 58.8 | 74.8 | 54.9 | 82.5 | 14.0 | 60.6 |
| MLP | 54.0 | 14.3 | 46.3 | 71.3 | 60.1 | 76.9 | 58.5 | 83.1 | 14.8 | 60.6 |
| CNN | 53.0 | 12.0 | 43.5 | 71.6 | 59.2 | 77.5 | 55.2 | 82.6 | 18.8 | 56.3 |
| BiLSTM | 50.8 | 0 | 44.9 | 71.3 | 59.4 | 78.0 | 54.0 | 81.0 | 12.0 | 56.3 |
| *— task-specific distillation (ours) CMOW not pretrained —* | | | | | | | | | | |
| Linear probe | 53.7 | 13.8 | 45.3 | 72.1 | 62.5 | 80.9 | 53.4 | 84.1 | 15.2 | 56.3 |
| MLP | 54.8 | 15.1 | 45.6 | 72.8 | 60.6 | 82.6 | 55.6 | 84.3 | 20.0 | 56.3 |
| CNN | 54.6 | 13.4 | 45.6 | 72.3 | 61.2 | 82.6 | 56.3 | **86.8** | 15.0 | 57.8 |
| BiLSTM | 53.2 | 16.7 | 44.9 | 72.1 | 64.8 | 80.6 | 54.2 | 82.9 | 7.9 | 54.9 |
| *— task-specific distillation (ours) CMOW pretrained —* | | | | | | | | | | |
| Linear probe | 54.3 | 20.8 | 48.6 | 71.3 | 60.3 | 78.4 | 54.9 | 84.5 | 13.8 | 56.3 |
| MLP | 55.4 | 18.9 | 50.4 | 72.3 | 61.3 | 79.3 | 55.2 | 83.0 | 17.9 | 60.6 |
| CNN | 56.2 | 18.3 | 50.1 | 71.8 | 60.5 | 80.6 | 57.0 | 85.0 | 13.2 | **69.0** |
| BiLSTM | 51.4 | 0 | 44.2 | 68.4 | 59.8 | 81.1 | 55.2 | 82.3 | 15.0 | 56.3 |
| *— task-specific distillation (ours) CMOW/CBOW-Hybrid not pretrained —* | | | | | | | | | | |
| Linear probe | 54.4 | 17.0 | 47.0 | 72.6 | 61.1 | 81.4 | 53.4 | 84.5 | 15.1 | 57.8 |
| MLP | 54.4 | 13.8 | 50.0 | 73.0 | 60.4 | 78.6 | 53.8 | 84.9 | 18.5 | 56.3 |
| CNN | 53.6 | 12.0 | 42.1 | 72.6 | 60.9 | 79.6 | 52.7 | 85.7 | 16.3 | 60.6 |
| BiLSTM | 52.4 | 0 | 43.2 | 72.1 | 61.2 | 80.0 | 57.4 | 83.0 | 18.1 | 56.3 |
| *— task-specific distillation (ours) CMOW/CBOW-Hybrid pretrained —* | | | | | | | | | | |
| Linear probe | 53.9 | 19.1 | 41.0 | 71.8 | 57.6 | 78.7 | 57.8 | 83.7 | 16.2 | 59.2 |
| MLP | 55.3 | 22.1 | 47.4 | 71.6 | 60.0 | 79.5 | 57.8 | 84.1 | 18.1 | 56.3 |
| CNN | 54.0 | 20.7 | 44.5 | 71.8 | 59.9 | 79.7 | 54.9 | 85.9 | 9.9 | 59.1 |
| BiLSTM | 53.7 | 17.0 | 40.6 | 71.8 | 61.3 | 80.3 | 57.4 | 82.5 | 14.0 | 59.2 |

Table 10: Scores on the GLUE development set with DiffCat two-sequence encoding

| Task-Specific Distillation | Score | CoLA | MNLI-m | MRPC | QNLI | QQP | RTE | SST-2 | STS-B | WNLI |
|---|---|---|---|---|---|---|---|---|---|---|
| *— task-specific finetuning (ours) —* | | | | | | | | | | |
| Teacher BERT-base | 78.9 | 57.9 | 84.2 | 84.6 | 91.4 | 89.7 | 67.9 | 91.7 | 88.0 | 54.9 |
| *— task-specific distillation (ours) CBOW not pretrained —* | | | | | | | | | | |
| Linear probe | 53.8 | 11.5 | 46.6 | 72.8 | 62.2 | 76.7 | 52.7 | 83.5 | 22.0 | 56.3 |
| MLP | 61.0 | 14.3 | 57.8 | 77.2 | 70.3 | 86.0 | 56.7 | 82.3 | 47.0 | **57.7** |
| CNN | 53.8 | 11.2 | 51.5 | 75.0 | 65.8 | 81.3 | 53.1 | 82.3 | 7.2 | 56.3 |
| BiLSTM | 48.4 | 11.5 | 31.8 | 68.3 | 66.8 | 63.2 | 56.7 | 83.5 | 1.5 | 56.3 |
| *— task-specific distillation (ours) CBOW pretrained —* | | | | | | | | | | |
| Linear probe | 56.3 | 9.0 | 47.1 | 72.8 | 64.8 | 77.1 | 53.4 | 82.5 | 43.4 | 56.3 |
| MLP | 63.8 | 14.0 | 61.7 | **78.2** | 70.8 | 86.2 | 57.4 | 83.8 | **66.0** | 56.3 |
| CNN | 53.7 | 10.9 | 55.0 | 73.8 | 66.2 | 82.1 | 53.1 | 82.2 | 3.8 | 56.3 |
| BiLSTM | 47.7 | 0 | 32.7 | 68.4 | 69.6 | 63.2 | 55.6 | 82.5 | 1.3 | 56.3 |
| *— task-specific distillation (ours) CMOW not pretrained —* | | | | | | | | | | |
| Linear probe | 55.1 | 10.9 | 54.3 | 71.8 | 62.7 | 80.9 | 56.0 | 85.2 | 17.6 | 56.3 |
| MLP | 63.2 | 14.2 | 61.9 | 75.5 | 72.4 | 86.3 | 55.2 | 83.7 | 62.7 | 56.3 |
| CNN | 55.4 | 12.4 | 45.3 | 72.3 | 61.5 | 82.6 | 57.4 | 84.3 | 26.1 | 56.3 |
| BiLSTM | 47.5 | 0 | 31.8 | 70.3 | 49.5 | 81.0 | 55.6 | 83.4 | 0 | 56.3 |
| *— task-specific distillation (ours) CMOW pretrained —* | | | | | | | | | | |
| Linear probe | 56.3 | 22.4 | 48.4 | 72.5 | 61.3 | 81.9 | 54.5 | 83.9 | 24.2 | **57.7** |
| MLP | 61.2 | 20.9 | 60.2 | 73.8 | 64.6 | 85.9 | 54.9 | 84.4 | 49.4 | 56.3 |
| CNN | 53.4 | 18.5 | 40.6 | 71.8 | 58.2 | 68.3 | 54.9 | 85.4 | 26.9 | 56.3 |
| BiLSTM | 49.7 | 0 | 32.7 | 68.3 | 67.2 | 82.9 | 57.0 | 82.5 | 0 | 56.3 |
| *— task-specific distillation (ours) Hybrid not pretrained —* | | | | | | | | | | |
| Linear probe | 51.7 | 11.2 | 39.0 | 71.1 | 49.5 | 81.8 | 56.0 | 85.2 | 14.3 | **57.7** |
| MLP | 62.5 | 13.1 | 62.5 | 74.3 | 71.5 | **86.6** | 58.1 | 83.1 | 58.6 | 56.3 |
| CNN | 52.8 | 11.9 | 45.3 | 71.6 | 61.4 | 84.8 | 55.2 | 85.4 | 2.9 | 56.3 |
| BiLSTM | 50.9 | 0 | 42.6 | 70.1 | 60.3 | 79.3 | 56.0 | 84.4 | 9.3 | 56.3 |
| *— task-specific distillation (ours) Hybrid pretrained —* | | | | | | | | | | |
| Linear probe | 54.0 | 19.6 | 45.7 | 71.3 | 63.4 | 80.9 | 54.2 | 84.1 | 11.7 | 54.9 |
| MLP | 62.7 | 20.9 | 62.6 | 74.5 | 68.6 | 85.7 | 56.3 | 83.1 | 56.2 | 56.3 |
| CNN | 57.9 | 19.6 | 37.6 | 75.7 | 62.0 | 85.4 | 54.9 | 82.3 | 48.5 | 54.9 |
| BiLSTM | 52.1 | 0 | 48.0 | 68.4 | 71.9 | 85.3 | 56.7 | 82.5 | 0 | 56.3 |

Table 11: Scores on the GLUE development set with DiffCat encoding and the bidirectional CMOW/CBOW-Hybrid model

| Task-Specific Distillation | Score | CoLA | MNLI-m | MRPC | QNLI | QQP | RTE | SST-2 | STS-B | WNLI |
|---|---|---|---|---|---|---|---|---|---|---|
| *— task-specific finetuning (ours) —* | | | | | | | | | | |
| Teacher BERT-base | 78.9 | 57.9 | 84.2 | 84.6 | 91.4 | 89.7 | 67.9 | 91.7 | 88.0 | 54.9 |
| *— task-specific distillation (ours) Bidirectional Hybrid, not pretrained —* | | | | | | | | | | |
| Linear probe | 53.5 | 11.6 | 39.4 | 71.6 | 64.3 | 82.5 | 56.3 | 85.0 | 14.6 | 56.3 |
| MLP | 63.2 | 13.0 | **63.3** | 75.7 | **72.6** | 86.1 | 57.4 | 83.3 | 59.7 | 57.7 |
| CNN | 52.7 | 14.5 | 37.3 | 71.3 | 60.8 | 86.4 | 55.2 | 85.8 | 6.6 | 56.3 |
| *— task-specific distillation (ours) Bidirectional Hybrid, pretrained —* | | | | | | | | | | |
| Linear probe | 55.5 | 18.1 | 42.4 | 72.1 | 64.9 | 81.2 | 56.7 | 85.2 | 22.5 | 56.3 |
| MLP | **64.6** | **23.3** | 61.8 | 75.0 | 72.0 | 86.3 | **59.9** | 82.9 | 62.9 | 57.7 |
| CNN | 55.1 | 20.5 | 39.3 | 73.8 | 61.3 | 85.9 | 56.3 | 85.5 | 15.9 | 57.7 |

Table 12: Number of parameters without DiffCat encoding

| | CoLA, MRPC, QNLI, QQP, SST-2, RTE, WNLI | MNLI | STS-B |
|---|---|---|---|
| — *task-specific distillation (ours) CBOW* — | | | |
| Linear probe | | 47,861,634 | 47,862,419 | 47,876,549 |
| *– only classifier* | | 3,138 | 3,923 | 18,053 |
| MLP | | 48,647,498 | 48,648,499 | 48,666,517 |
| *– only classifier* | | 789,002 | 790,003 | 808,021 |
| CNN | | 47,862,708 | 47,864,737 | 47,901,259 |
| *– only classifier* | | 4,212 | 6,241 | 42,763 |
| BiLSTM | | 53,704,002 | 53,705,027 | 53,723,477 |
| *– only classifier* | | 5,845,506 | 5,846,531 | 5,864,981 |
| — *task-specific distillation (ours) CMOW* — | | | |
| Linear probe | | 23,932,386 | 23,933,171 | 23,947,301 |
| *– only classifier* | | 3,138 | 3,923 | 18,053 |
| MLP | | 24,718,250 | 24,719,251 | 24,737,269 |
| *– only classifier* | | 789,002 | 790,003 | 808,021 |
| CNN | | 23,933,460 | 23,935,489 | 23,972,011 |
| *– only classifier* | | 4,212 | 6,241 | 42,763 |
| BiLSTM | | 24,853,978 | 24,854,371 | 35,022,869 |
| *– only classifier* | | 924,730 | 925,123 | 110,936,21 |
| — *task-specific distillation (ours) Hybrid* — | | | |
| Linear probe | | 24,420,802 | 24,421,603 | 24,436,021 |
| *– only classifier* | | 3,202 | 4,003 | 18,421 |
| MLP | | 25,222,602 | 25,223,603 | 25,241,621 |
| *– only classifier* | | 805,002 | 806,003 | 824,021 |
| CNN | | 24,420,558 | 24,421,855 | 24,445,201 |
| *– only classifier* | | 2,958 | 4,255 | 27,601 |
| BiLSTM | | 30,328,642 | 30,329,667 | 30,348,117 |
| *– only classifier* | | 5,911,042 | 5,912,067 | 5,930,517 |

Table 13: Number of parameters with DiffCat two-sequence encoding

| | CoLA, MRPC, QNLI, QQP, SST-2, RTE, WNLI | MNLI | STS-B |
|---|---|---|---|
| *— task-specific distillation (ours) CBOW —* | | | |
| Linear probe | 47,867,906 | 47,870,259 | 47,912,613 |
| *– only classifier* | 9,410 | 11,763 | 54,117 |
| MLP | 50,215,498 | 50,216,499 | 50,234,517 |
| *– only classifier* | 2,357,002 | 2,358,003 | 2,376,021 |
| CNN | 47,865,932 | 47,869,313 | 47,930,171 |
| *– only classifier* | 7,436 | 10,817 | 71,675 |
| BiLSTM | 147,477,458 | 147,479,811 | 147,522,165 |
| *– only classifier* | 99,618,962 | 99,621,315 | 99,663,669 |
| *— task-specific distillation (ours) CMOW —* | | | |
| Linear probe | 23,938,658 | 23,941,011 | 23,983,365 |
| *– only classifier* | 9,410 | 11,763 | 54,117 |
| MLP | 26,286,250 | 26,287,251 | 26,305,269 |
| *– only classifier* | 2,357,002 | 2,358,003 | 2,376,021 |
| CNN | 23,936,684 | 23,940,065 | 24,000,923 |
| *– only classifier* | 7,436 | 10,817 | 71,675 |
| BiLSTM | 123,548,210 | 123,550,563 | 123,592,917 |
| *– only classifier* | 99,618,962 | 99,621,315 | 99,663,669 |
| *— task-specific distillation (ours) Hybrid —* | | | |
| Linear probe | 24,427,202 | 24,429,603 | 24,472,821 |
| *– only classifier* | 9,602 | 12,003 | 55,221 |
| MLP | 26,822,602 | 26,823,603 | 26,841,621 |
| *– only classifier* | 2,405,002 | 2,406,003 | 2,424,021 |
| CNN | 24,424,990 | 24,427,583 | 24,474,257 |
| *– only classifier* | 7,390 | 9,983 | 56,657 |
| BiLSTM | 128,143,202 | 128,145,603 | 128,188,821 |
| *– only classifier* | 103,725,602 | 103,728,003 | 103,771,221 |
| *— task-specific distillation (ours) Hybrid bidirectional —* | | | |
| Linear probe | 36,640,802 | 36,644,403 | 36,709,221 |
| *– only classifier* | 14402 | 18003 | 82,821 |
| MLP | 40,231,402 | 40,232,403 | 4,025,0421 |
| *– only classifier* | 3,605,002 | 3,606,003 | 3,624,021 |
| CNN | 36,638,164 | 36,641,729 | 36,705,899 |
| *– only classifier* | 11,764 | 15,329 | 79,499 |
| BiLSTM | 451,437,602 | | 451,528,821 |
| *– only classifier* | 414,811,202 | | 414,902,421 |

