# OpenReview forum: "Cross-Architecture Distillation Using Bidirectional CMOW Embeddings"
_ICLR.cc/2022/Conference — ICLR 2022 Submitted_

### Official Review · Reviewer_AXE2 · 2021-11-01

**Correctness:** 3
**Technical Novelty And Significance:** 3
**Empirical Novelty And Significance:** 3
**Recommendation:** 5
**Confidence:** 3

**Main Review:**

I am curious how it performs with a single CBOW or a single CMOW, not a hybrid one.

Did you try other sizes of matrix embeddings of size? How does it perform? Should the CBOW and CMOW embeddings have the same dimension size? What are their individual functions for CBOW and CMOW? Are they complementary or just redundant? I know CMOW is from another paper, but as a reader, we might want to read a self-contained paper.



**Summary Of The Paper:**

The paper proposes to distill a BERT model to a smaller one,  a CMOW/CBOW-Hybrid. Its performance is better than ELMo, but worse than the typical distillation method using transformer-based students like DistillBERT. The authors claim that CMOW/CBOW-Hybrid is much faster. The paper is clearly written.

**Summary Of The Review:**


To distill a better model to a weaker model, one can try different student models, either using shallower/narrower Transformer-based models or other models. If we could tolerate a big performance drop (as in this paper, the authors were satisfied that it exceeds ELMo), there are much more choices, of which CMOW/CBOW-Hybrid might be a good one. My concern is that using CMOW/CBOW-Hybrid is not that interesting, one might also try CBOW with convolutions, for example. It may help if the authors could provide more insights to choose CMOW/CBOW-Hybrid,  or why the investigation of  CMOW/CBOW-Hybrid is important, as we might have so many lightweight models as the student model.

---

> ### Author Response · Authors · 2021-11-18
> **Author Response and Change Summary**
>
> Thank you for your thoughtful review.
>
> The original CMOW paper (Mai, 2019) has extensively analyzed the CMOW and CBOW components individually and found that joint training is generally preferable. It is not necessary that they have the same dimension size. We have added this to the paper (Section 2).
>
> Certainly, there are more choices for efficient student models. Wasserblatt (2020) has used a feed forward net on a bag-of-words as well as an LSTM, both of which are outperformed by ours on the tasks that they report (revisit Table 4).
>
> We have also experimented with other embedding models (pure CMOW and pure CBOW) and other downstream classifiers (like 2D-CNN on CMOW), for which we report numbers in the appendix. Table 7 shows the best-performing methods per task. We see that CMOW+CNN and CBOW+MLP do have an edge over CMOW/CBOW-Hybrid in some tasks, but CMOW/CBOW-Hybrid performed best on average, which is why we have sticked to this model for the main paper.
>
> **Changes**:
>
> - Added more background information for the CMOW and CBOW components to Section 2 (Methods)

---

> > ### Comment · Reviewer_AXE2 · 2021-11-26
> > **Thanks for your reply**
> >
> > Thanks for your reply.
> >
> > Other than performance, do you have any other insights for the importance to investigate of CMOW/CBOW-Hybrid is important, e.g., compared to other lightweight models? Why should CMOW/CBOW-Hybrid be the model to distill PLMs and how can the community benefit from CMOW/CBOW-Hybrid? It seems that CMOW is computationally expensive for long documents because it involves a product of a sequence of matrices.
> >
> > Also, I still did not get why CMOW should always be combined with a CBOW. What information do we lose if we remove CBOW in CMOW/CBOW-Hybrid? This is not about performance, it is about the design itself. It is risky if we could see more papers that use A/B Hybrid but do not explain the complementarity between A and B.
> >
> > By tolerancing some performance drops, we could have many more combinations between A/B/C/D as a student model to distill PLMs. However, one has to explain why such a new student model (especially a hybrid one) matters in our community.
> >
> > Sorry that some of the comments may be harsh, I am happy if you could correct me.

---

> > > ### Author Response · Authors · 2021-11-26
> > > **Author Response**
> > >
> > >
> > > Thank you very much for your additional comments.
> > >
> > > **Computational Costs for CMOW** The *main advantage of CMOW* is actually that, due to its associative property, CMOW only requires O(log L) *sequential* steps to encode a sequence of length L (think of a binary tree on-top of the sequence). For the same reason, also frequent n-grams can be pre-computed in a dynamic programming fashion, such that practitioners can arbitrarily trade-off speed and space requirements. Thus, it is efficient while being sensitive to word order. It fills a gap between RNNs/LSTMs, which require O(L) sequential steps, and Transformers, which need to compute O(L^2) attention weights (at least in their standard formulation).
> > >
> > > **Importance of CMOW/CBOW-Hybrid** *Why combining CMOW with CBOW* is a question answered in the original paper: In controlled experiments with same dimensionality, the CMOW paper shows that both CMOW and CBOW have their individual strengths, and a combined hybrid version is superior over using a single one of them and thus should be used. The rationale is that memorizing Word Content (which words are in a sequence) is most correlated with downstream performance (see Conneau 2018's "what you can cram into a single vector") and that this is easier with summation rather than matmul.
> > >
> > > We are aware that, in theory, even vector addition could be modeled as matrix multiplication. Still, to model N-dim vector addition with matrices, one would need N-square matrices with mostly zeros. On the other hand, vector summation does not harm the associative property (main advantage above) and its optimization is easier.
> > >
> > > **Other student models** Undoubtedly, many other options for student models are possible. The most straightforward would probably be CBOW+MLP (Wasserblatt 2020, and our own experiments in the appendix). But as our experiments show, word order (modeled by CMOW) is helpful for most of the GLUE tasks, and in the end, leads to a higher average score. We believe that our results open up a whole new range of future work. For instance, also CMOW only + a 2D-CNN on the embedding matrix has achieved very good scores on some of the tasks (even without pretraining). Thus, we suggest to further investigate distillation in CMOW-style models.
> > >
> > > **We will add these detailed explanations to the paper in a camera-ready version.**

---

> > > > ### Comment · Reviewer_AXE2 · 2021-11-29
> > > > **Thanks for your replies.**
> > > >
> > > > I appreciate the provided replies.
> > > >
> > > > Ｉ did not get the following statement.
> > > > > The main advantage of CMOW is actually that, due to its associative property, CMOW only requires O(log L) sequential steps to encode a sequence of length L (think of a binary tree on-top of the sequence)
> > > >
> > > > Why does it require O(log L) sequential steps?  It seems with a $L$-length sequence, it needs $L-1$ matrix multiplications
> > > >
> > > > Also, how
> > > > > it fills a gap between RNNs/LSTMs and Transformers
> > > >
> > > > Could you please provide more details about it.

---

> > > > > ### Author Response · Authors · 2021-11-29
> > > > > **Author Response**
> > > > >
> > > > > Thank you for your interest in our work.
> > > > >
> > > > > With CMOW and also CMOW/CBOW-Hybrid, we exploit the fact that matrix multiplication is associative, i.e., it holds $(A \cdot B) \cdot C = A \cdot (B \cdot C)$. For a sequence of matrices $A_1, \ldots , A_L$, we can first compute $X_1 = A_1 \cdot A_2$, and $X_2 = A_3 \cdot A_4$, ..., followed by multiplying $X_1 \cdot X_2 \cdot \ldots$, and so on. Thus, the parts $A_i \cdot A_{i+1}$ and $A_{i+2} \cdot A_{i+3}$ do not depend on each other and can be computed in parallel.  This holds for both pure CMOW and CMOW/CBOW-Hybrid and reduces the number of sequential matrix multiplications.
> > > > >
> > > > > Because all inputs needs to be considered, the asymptotic complexity is still $\mathcal{O}(L)$, but only $\mathcal{O}(\log L)$ sequential steps (that depend on the output of each other), are necessary to encode a sequence of length L. As such, the asymptotic complexity for encoding a sequence is different from other sequence models like RNN/LSTM and Transformer but word order is still reflected in the model, in contrast to bag-of-word-based models that discard word order altogether.
> > > > >
> > > > > Hoping this clarifies our statements from above.

---

> > > > > > ### Comment · Reviewer_AXE2 · 2021-11-29
> > > > > > **the way for complexity**
> > > > > >
> > > > > > In the case when we consider the product between *two* matrices in a single sequential step, it results in $\mathcal{O} (log_2 L)$ sequential steps. But when we calculate the product between *N* matrices in a single sequential step,  it results in $\mathcal{O} (log_N L)$ sequential steps. From this perspective, $\mathcal{O} (log L)$ seems to make sense.
> > > > > >
> > > > > > My concern is that, could we say, Transformer only needs $\mathcal{O} (1)$  sequential steps to process a $L$-length document/sentence? This is because, in a single sequential step, Transformer could see all tokens in a document/sentence. This involves two indicators that might be important:
> > > > > > -  (a) the calculation granularity in a single sequential step, e.g., how many tokens are visible in a single sequential step.
> > > > > > -  (b) how many sequential steps to be needed to model the overall representation in a document/sentence.
> > > > > >
> > > > > > We have to trade off a) and b), it seems **a smaller (a) indicates a bigger (b)**, and **a smaller (b) indicates a bigger (a)**. From this perspective, RNN is the former while Transformer is the latter.
> > > > > >
> > > > > > Do you agree with the statement above? and how do you rethink the benefit of CMOW from parallel computing?
> > > > > >
> > > > > > Correct me if I am wrong.

---

> > > > > > > ### Author Response · Authors · 2021-11-30
> > > > > > > **Author Response**
> > > > > > >
> > > > > > > Thank you for going so much into the details. We pretty much agree with the statements above.
> > > > > > >
> > > > > > > Transformers have $\mathcal{O}(L^2)$ compute effort (the attention scores), and $\mathcal{O}(1)$ sequential steps.
> > > > > > >
> > > > > > > CMOW/CBOW-Hybrid have $\mathcal{O}(L)$ compute effort and $\mathcal{O}(\log L)$ sequential steps.
> > > > > > >
> > > > > > > So yes, it is sort of a trade-off.
> > > > > > > In Table 5, we have reported the observed inference speed of CMOW/CBOW, BERT-base, TinyBERT, and others. In classic BERT-base, an $\mathcal{O}(1)$ inference step is six times slower than our CMOW/CBOW-Hybrid and TinyBERT.
> > > > > > >
> > > > > > > In terms of parallel computing, one further benefit of CMOW-based models is that, e.g., n-grams can be precomputed and stored in look up tables. This allows practioners to trade-off speed vs space requirements.

---

### Official Review · Reviewer_M3tk · 2021-11-02

**Correctness:** 2
**Technical Novelty And Significance:** 1
**Empirical Novelty And Significance:** Not applicable
**Recommendation:** 3
**Confidence:** 5

**Main Review:**

The main contribution of this work mainly lies in the bidirectional CMOW and a two-stage knowledge distillation method. However, the technical contribution in bidirectional CMOW is quite limited, the related extension has been widely used in existing works, like bidirectional LSTM. Furthermore, the two-stage knowledge distillation method was originally proposed in TinyBERT, a well-known work on BERT distillation, which is directly used in this work. The proposed method has poor performances on several GLUE tasks, e.g., CoLA and MRPC, and many related baselines are not included, e.g. TinyBERT, BERT-PKD, MiniLM. The proposed method even does not have advantages on #parameters and inference time with comparison to 4-layer TinyBERT.

**Summary Of The Paper:**

This paper presents a knowledge distillation method for BERT, where BERT-base acts as a teacher network and  CMOW is used as a student network. To get a relatively strong student network, the authors extend the vanilla CMOW to bidirectional CMOW. The authors also adopt a two-stage distillation method consisting of general distillation and task-specific distillation to further improve the performances. Finally, the authors evaluate the proposed method on GLUE dataset.

**Summary Of The Review:**

The bidirectional extension of CMOW is straightforward, and the two-stage knowledge distillation method is originally  proposed by TinyBERT. The final performances of the proposed method performs worse than most baselines, e.g., TinyBERT, DistilBERT, MiniLM, BERT-PKD.

---

> ### Author Response · Authors · 2021-11-18
> **Author Response and Change Summary**
>
> Thank you for your thoughtful review. Please allow us to clarify the comparison with approaches like BERT-PKD/TinyBERT:
>
> First, we did not intend to suggest that the two-stage procedure is novel.
>
> Our point is this: One has to distinguish between general distillation and task-specific distillation. These are conceptually different. General distillation methods (e.g., DistilBERT or MobileBERT) do not need the teacher signal when fine-tuning for a new task. For task-specific distillation approaches such as TinyBERT (and BERT-PKD) it is necessary to keep the teacher model for fine-tuning for every task, which is *not* necessary for our approach. This has motivated us to analyze how the two approaches general  distillitation vs task-specific distillation compare with CMOW/CBOW-Hybrid.
>
> The result is that general distillation is sufficient for CMOW/CBOW-Hybrid, which is a clear advantage as discussed above. But we acknowledge that TinyBERT achieves a higher performance on GLUE. Our approach is not to be considered the end of the path, but a next step. So, yes, there is still some future work necessary to improve the performance on CoLA with efficient student models.
>
> But, we believe that we have made a notable step into that direction as we do outperform other cross-architecture distillation approaches (Wasserblatt 2020's CBOW-FFN and BiLSTM student models) by a large margin (for instance: 23.3 vs 10.0 on CoLA).
>
> **Changes**:
>
> - We have added TinyBERT to the model size and throughput comparison in Section 4 (Results)
> - We have added a more detailed reflection how our approach compares to TinyBERT in Section 5 (Discussion)

---

> > ### Comment · Reviewer_M3tk · 2021-11-28
> > **Thanks for your reply**
> >
> > Thanks for your reply and efforts!
> >
> > On the definition of general distillation, DistilBERT or MobileBERT can be attributed to task-agnostic distillation, and they only need to do KD during the pre-training stage and can be directly fine-tuned on downstream tasks.
> >
> > On the performance on CoLA task, I agree that the proposed method outperforms other baselines (23.3 vs 10.0 on CoLA), however, there still exists a big gap to the performance of BERT-base, even to some weak KD method, like DistilBERT4.
> >
> > In the Table 5, tinybert and the proposed method have the same encoding speed 30.0k (about 6.5x faster than BERT-base), while in the original tinybert paper, the inference time of tinybert4 is about 9.4x faster than BERT-base.

---

### Official Review · Reviewer_mHTJ · 2021-11-03

**Correctness:** 4
**Technical Novelty And Significance:** 3
**Empirical Novelty And Significance:** 2
**Recommendation:** 6
**Confidence:** 4

**Main Review:**

Strengths:
1. The proposed idea is novel: the CMOW model can only be used in sentence-level tasks. The authors extend CMOW using the bidirectional variant to extract token-level representations, so now it can be used for the masked language modeling training objective.
2. DiffCat is designed specifically for two-sequence tasks. It improves the performance by 32% compared to naive joint encoding.
3. The proposed method can achieve comparable results to ELMo, while having a much higher encoding speed.

Weaknesses:
1. The evaluation of the proposed method is focused on the GLUE task, which is a collection of classification tasks. I encourage the authors to test the model on other tasks as well. For example, DistilBERT can also perform well on question answering datasets like SQuAD. It's interesting to see whether this shallow model can perform appropriately on more complex tasks.
2. How can we trade-off between this fast inference speed and good performance? It seems that this model is very fast at inference speed, but the performance was compromised. It would be better if we are allowed to find a balance between speed and performance according to our demand. For example, TinyBERT provides a 4-layer version and a 6-layer version, so we can find a balance between the inference time and performance. In this paper, it seems that the author only provides 20x20 matrix embeddings for the CMOW model.

**Summary Of The Paper:**

The authors of this paper try to distill large pretrained language models into a bidirectional CMOW/CBOW-Hybrid model. The proposed architecture was designed to output per-token representations, so as to perform knowledge distillation on the masked language modeling pretraining task. A two-sequence encoding scheme (DiffCat) was designed for the downstream task on sentence pairs. Empirical results show that the proposed method can achieve comparable results to ELMo while using only half of the parameters and providing three times faster inference speed.

**Summary Of The Review:**

The paper presents a novel and useful idea: distill a complex pretraining model into a simple CMOW model. The initial result is promising while some limitations can be further explored. I think this paper is worth accepting because it can motivate the research on smaller distilled PreLMs.

---

> ### Author Response · Authors · 2021-11-18
> **Author Response and Change Summary**
>
> Thank you for your thoughtful review.
>
> We agree that evaluating CMOW-Hybrid on even more challenging tasks such as question answering on SQuAD would be interesting. To some extent, we have laid the foundation for using CMOW-style models on SQUAD by introducing the token-level representations that would also be necessary to predict spans in the context of question answering. For now, we leave it as future work.
>
> Likewise, having a dedicated comparison of different embedding sizes would be indeed desirable. For now, we have focused on evaluating different downstream classifier models while keeping the embedding size fixed: Linear, MLP, 2D-Convolution on CMOW representation, as well as using an LSTM to carry out the pooling of the sequence. We expect that the scores would improve with higher dimensional embeddings because of general arguments that even a random projection into higher dimensions leads to improvements in downstream performance (Wieting & Kiela, 2019, https://arxiv.org/abs/1901.10444).
>
>
> **Changes to the paper**
>
> - Mentioned question answering in the future work paragraph and sketched how our per-token CMOW representations would help with that
> - Added a discussion of the trade-off between size and performance to Section 5 (Discussion) under 'Limitations'

---

> > ### Comment · Reviewer_mHTJ · 2021-11-29
> > **Thanks for your reply**
> >
> > Thanks for your reply and the changes to the paper. I will keep my score unchanged.

---

### Decision · Program_Chairs · 2022-01-20

**Decision:**

Reject

**Comment:**

This paper presents a method for distilling pretrained models (such as BERT) into a different student architecture (CMOW), and extend the CMOW architecture with a bidirectional component.  On a couple of datasets, results are comparable to DistilBERT a previous baseline. This paper is nice, but can be stronger with more empirical experiments on non-GLUE tasks (TriviaQA, Natural Questions, SQUAD for example).  Furthermore, I agree with Reviewer M3tk that there are many empirical comparisons with baselines such as TinyBERT missing and the argument of not needing the teacher model to be super convincing.